# An AUV-Aided Cross-Layer Mobile Data Gathering Protocol for Underwater Sensor Networks [note 1]

**DOI:** 10.3390/s20174813

**Published:** 2020-08-26

**Authors:** Faisal Abdulaziz Alfouzan, Seyed Mohammad Ghoreyshi, Alireza Shahrabi, Mahsa Sadeghi Ghahroudi

**Affiliations:** 1Department of Criminal Evidence, Naif Arab University for Security Sciences (NAUSS), College of Criminal Justice, Riyadh 14812, Saudi Arabia; 2School of Mathematical Sciences, University of Southampton, Southampton SO17 1BJ, UK; S.M.Ghoreyshi@soton.ac.uk; 3School of Computing, Engineering and Built Environment, Department Cyber Security and Networks (CSN), Glasgow Caledonian University, Glasgow G4 0BA, UK; A.Shahrabi@gcu.ac.uk (A.S.); Mahsa.Sadeghi@gcu.ac.uk (M.S.G.)

**Keywords:** underwater sensor networks, cross-layer routing, MAC algorithms, distributed clustering approach, autonomous underwater vehicles

## Abstract

Underwater sensor networks (UWSNs) have recently attracted much attention due to their ability to discover and monitor the aquatic environment. However, acoustic communication has posed some significant challenges, such as high propagation delay, low available bandwidth, and high bit error rate. Therefore, proposing a cross-layer protocol is of high importance to the field to integrate different communication functionalities (i.e, an interaction between data link layer and network layer) to interact in a more reliable and flexible manner to overcome the consequences of applying acoustic signals. In this paper, a novel Cross-Layer Mobile Data gathering (CLMD) scheme for Underwater Sensor Networks (UWSNs) is presented to improve the performance by providing the interaction between the MAC and routing layers. In CLMD, an Autonomous Underwater Vehicle (AUV) is used to periodically visit a group of clusters which are responsible for data collection from members. The communications are managed by using a distributed cross-layer solution to enhance network performance in terms of packet delivery and energy saving. The cluster heads are replaced with other candidate members at the end of each operational phase to prolong the network lifetime. The effectiveness of CLMD is verified through an extensive simulation study which reveals the performance improvement in the energy-saving, network lifetime, and packet delivery ratio with varying number of nodes. The effects of MAC protocols are also studied by studying the network performance under various MAC protocols in terms of packet delivery ratio, goodput, and energy consumption with varying density of nodes.

## 1. Introduction

A wide range of aquatic applications, such as pollution monitoring, military, intrusion detection, disaster prevention, and resource exploration, are now supported by the Underwater Sensor Networks (UWSNs) [1]. UWSNs face different challenges such as transmission reliability, low bit rate, sensor movement, limited bandwidth, and high latency [2,3,4].

A UWSN is an ad hoc network consisting of a group of acoustic sensors distributed within the marine environment for data collection. Some networks only use stationary sinks for data collection while some also take advantage of the mobile sinks (e.g., autonomous underwater vehicle (AUV)) [5]. In a UWSN with a stationary sink, sensors nodes close to the sink will die out sooner as they relay a high number of data packets [6]. This can affect network connectivity if the operation needs to continue for a longer time. Using a mobile sink can reduce energy consumption as well as limit the number of transmissions [7].

In an AUV tour, data packets are constantly collected from cluster heads (CHs) and uploaded into the static sink at the end of each tour. The network lifetime is extended as the packet transmission is limited to fewer hops (e.g., intracluster communications).

The channel communications are managed by using a Medium Access Control (MAC) protocol to share the medium among several sensors. Collisions and retransmissions are avoided by proper scheduling. Reliable transmissions are guaranteed by conflict management for sensors which provide fairness, low channel access delays, high throughput, and energy efficiency [8].

A distributed cross-layer solution is proposed to combine the functionality of routing and MAC layer while most of the works have only focused on a specific layer without any consideration for another one [9]. Cross-layer communication solutions have proven to be more efficient for resource management (e.g., bandwidth ) and energy-saving in UWSNs [10,11].

To the best of our knowledge, this work is the first cross-layer mobile data gathering protocol which has integrated the network and data link layers for three-dimensional (3D) UWSNs [12]. In this work, sensors are clustered within one-hop distance to their cluster head (CH) and an AUV can freely move through CHs for data collection. Proposed simulation results show significant improvement in the network performance.

The remainder of this paper is organized as follows. Section 2 introduces the related works. Section 3 provides a detailed description of the system model. Section 4 describes Cross-Layer Mobile Data gathering (CLMD) protocol in detail. Section 5 presents and discusses the results of our simulation study. Finally, Section 6 concludes the paper.

## 2. Related Work

The underwater environment has its own physical constraints and unique characteristics which should be taken into account during the design of the underwater routing and Medium Access Control (MAC) protocols over an aquatic network. These include slow propagation speed, limited bandwidth, energy constraints, node mobility with water current, and high deployment costs. UWSNs are significantly different from terrestrial (radio) sensor networks as they utilize acoustic waves with comparatively lower loss and longer range in the underwater environment than electromagnetic and optical waves [13,14,15,16,17].

Recently, a cross-layer protocol has increasingly been growing in network design due to its better performance [18,19]. Some variances based on the cross-layer group have been proposed.

An AUV in an AUV-aided Energy-Efficient Routing Protocol (AEERP) [20] travels in a planned elliptical trajectory in each data gathering process. Data packets are received and delivered to an AUV by selected sensors as the gateway. The gateway sensors selection is based on the residual energy and the distance to the AUV trajectory. Using a Shortest Path Tree (SPT) every member is assigned to a gateway. However, no bound on the hop distance from members to a gateway causes higher energy consumption. Moreover, a group of nodes may be isolated and unable to reach any gateway near the AUV trajectory that constitutes the void problem. Furthermore, there is no recovery mechanism for the dropped packets that are generated by these nodes.

In another mobile data gathering scheme AUV_PN [21], the network is divided into a number cluster, which is subdivided into several subcluster. To gather information from members, subclusters have a Path Node (PN). In every data-gathering operation, the AUV visits the nearest CH to regain the list of PNs and afterwards travels to every PN to obtain the data. This procedure repeat for next nearest CH after all PNs are visited in a cluster until all clusters are visited. The AUV returns to the base stations to pass all the aggregated data. Although the base station has received all the information, the constructed tour by AUV overpasses over itself and is not optimal. Moreover, in the larger network, the higher distance between PNs and members requires a higher power for transmission. Besides, network partitioning is also a complicated and energy-consuming process.

In a Cluster-based Mobile Data Gathering scheme (CMDG) [22], a group of sensors is selected as cluster heads in a distributed manner to collect data from other sensors. Then, an AUV tour is planned to collect these data from cluster heads with minimum latency. Although CMDG is effective in terms of the network layer, it still suffers from a lack of capability in terms of the MAC layer. The problem is that the network and MAC layer are not coordinated to increase the performance.

## 3. System Model

This section presents the network architecture and describes the acoustic propagation model in detail.

### 3.1. Network Architecture

A homogeneous network of sensors is assumed in terms of transmission range and power. Sensors are anchored to the subsea-floor to be fixed within the network topology [21,23]. To measure the pairwise distances, the Received Signal Strength Indicator (RSSI) is used by sensors. No localization to obtain the exact geographical coordinates is required in our proposed model.

An AUV can operate at a fixed depth above the acoustic sensors while it can move freely in all directions. The full geographical coordinates of each node can be obtained by the AUV when it approaches the sensors within the field. AUV can mark the locations where it receives data from [24]. All collected data by AUV will be delivered to a static sink on the water surface. These assumptions have been widely used in the literature [21,23,24].

### 3.2. Acoustic Propagation Model

In order to capture the underwater physical layer features, the BELLHOP ray tracing model [25] is used in our system model, which provides a propagation model similar to experimental implementation for UWSNs [26]. This channel model can consider absorbing boundary conditions to perform acoustic ray tracing for a given speed of sound profile. The ray trajectories of arrivals is predicted by using Gaussian beam tracing, taking into account parameters such as bathymetric information, the horizontal distance between sensors, the depth of source and receiver, signal frequency, and more importantly the sound speed profile [27].

To implement the ray tracing and transmission loss, the parameters are adopted from in [28]. The iso-speed sound velocity profile is used in our channel model with the bottom sound speed and the speed of sound in water equal to 1800 m/s and 1500 m/s, respectively. Furthermore, the sea bottom density is assumed as 1843 kg/m^3^, while the seawater density is taken as 1024 kg/m^3^.

## 4. Description of Our Proposed Cross-Layer Mobile Data-Gathering Protocol

This section briefly gives an overview of our proposed protocol followed by a description of each operational phase in detail.

### 4.1. Overview

The purpose of our proposed cross-layer protocol is to enhance the communication efficiency as well as to reduce the energy consumption by integrating functionalities of two layers in a distributed manner. To this end, CLMD allows the data packets to be efficiently delivered to the destination by sharing the underwater acoustic medium and aggregating data packets in a group of cluster heads to be visited by an AUV in the periodic tours.

CLMD includes four phases to operate: neighbor discovery, distributed clustering, AUV discovery, and normal operational phase, as illustrated in Figure 1. The beginning time of each phase has been set for all sensors in order to start and end together. A guard time has also been applied to avoid the effect of clock drafts that may occur over a long period of time.

In CLMD, in a distributed manner, a subset of sensors is selected as Cluster Heads (CHs) based on their priority considering the number of covering sensors, the closeness to the sink, and the residual energy of the sensor. Afterwards, an AUV tour is planned to visit CHs in a certain order for data collection.

The tour length should be minimized to reduce the end-to-end delay. The cluster heads are also reselected at each operational phase to prolong the network lifetime. A node with higher residual energy has a high chance to be selected as CH at the clustering phase.

### 4.2. Neighbor Discovery Phase

Sensors exchange a few short control packets to discover one-hop neighboring nodes and update the neighboring table, $N_{t}$. Each sensor selects its transmission time randomly within the discovery phase interval which is set to a predefined constant value. The discovery phase should be long enough to allow tables formation. This phase is fully explained in our previous works [6,29,30].

### 4.3. Distributed Clustering Phase

In order to determine the cluster heads; CHs; and their members, CMs, we use the same procedure that used in [6,29,30].

In this model, we extend our previous work by integrating the routing protocol in [6] with an efficient MAC protocol [29,30] which can enhance the mobile data-gathering efficiency. In this cross-layer design, we take advantage of the AUV discovery phase to assign the time frames to each cluster head, and subsequently assign time slots to each member of the cluster by CH. In our cross-layer mobile data-gathering protocol, a cluster head can be determined based on the priority value of a sensor node, which depends on a sensor degrees (one-hop neighboring degrees), distance (from a sensor to the sink), and energy. The priority value, *P*, of each sensor at time step *t* can be calculated by using the following equation,
(1)$$P\left(t,i\right)=\alpha \left(N_{Deg}\left(i\right)\right)+\beta \left(N_{Dis}\left(i\right)\right)+\gamma \left(N_{Ex}\left(t,i\right)\right),$$
where *t* is the current time step and *i* is a unique identifier (ID) of a sensor node. α, β, and γ are the degree weight, distance weight, and expected energy weight of a sensor node, respectively, which all of them should be equal to 1 (α + β + γ = 1). $N_{Deg}\left(i\right)$ is a normalized degrees of each node which can be obtained by ($\frac{Degree(i)}{MaxDegree(\Delta )}$) and $N_{Dis}\left(i\right)$ is a normalized distance to the sink which can be given by ($1-(\frac{Distance(i)}{MaxDistance(\Delta )})$). The normalized expected energy, $N_{Ex}\left(\lambda ,i\right)$, can be calculated by using the following equation,
(2)$$N_{Ex}\left(t,i\right)=\left(1-log\left(\frac{9\times E_{i}\left(t+1\right)}{InitialEnergy}+1\right)\right),$$
where $E_{i}\left(t+1\right)$ denotes the expected consumed energy at time step t+1.

#### Expected Consumed Energy Calculation

The sensor set can be represented as $S=\{s_{1},s_{2},s_{3},...,s_{n}\}$ while $s_{i}$ is a node in the network. We have the following definition.
$\begin{array}{c} \forall s_{i}\in S,\;\;\;\;\;\;\;\;\;\;\;\;\;\;\;\;\delta =\lambda_{i}+\Delta_{i} \end{array}$(3)$\begin{array}{c} \mathrm{where} \end{array}$(4)$\begin{array}{c} \delta =\{t_{0},t_{1},t_{2},...,t_{n}\}\;\mathrm{represents}\;\mathrm{the}\;\mathrm{time}\;\mathrm{steps} \end{array}$(5)$\begin{array}{c} \Delta_{i}=\;\mathrm{Set}\;\mathrm{of}\;\mathrm{time}\;\mathrm{steps}\;\mathrm{that}\;s_{i}\;\mathrm{was}\;\mathrm{the}\;\mathrm{member} \end{array}$(6)$\begin{array}{c} \lambda_{i}=\;\mathrm{Set}\;\mathrm{of}\;\mathrm{time}\;\mathrm{steps}\;\mathrm{that}\;s_{i}\;\mathrm{was}\;\mathrm{CH} \end{array}$(7)


For instance, if $\Delta_{i}=\left\{t_{2},t_{4}\right\}$ and $\lambda_{i}=\left\{t_{1},t_{3}\right\}$, it means that $s_{i}$ has been a CH at $t_{1}$, $t_{3}$ and has been a member at $t_{2}$, $t_{4}$.

The expected consumed energy of each node at t+1 can be calculated as
(8)$$E_{i}\left(t+1\right)=E_{\lambda_{i}}\left(t+1\right)+E_{\Delta_{i}}\left(t+1\right)+E_{\lambda_{i_{CH\_AUV}}}\left(t+1\right)$$
where $E_{\lambda_{i}}\left(t+1\right)$ is the expected consumed energy as a CH at t+1 and so far, $E_{\Delta_{i}}\left(t+1\right)$ is the expected consumed energy as a member so far, and $E_{\lambda_{i_{CH\_AUV}}}\left(t+1\right)$ is the expected consumed energy in the CH and AUV communication at t+1 and so far. We have the following equations,
(9)$$E_{\Delta_{i}}\left(t+1\right)=\left|\Delta \right|\times OP\times \phi \times E_{tran}$$
(10)$$E_{\lambda_{i}}\left(t+1\right)=\left(\sum_{N\in \lambda_{i}}D_{i}\left(N\right)+D_{i}\left(t+1\right)\right)\times \times OP\times \phi \times E_{rec}$$
(11)$$E_{\lambda_{i_{CH\_AUV}}}\left(t+1\right)=\left(\sum_{N\in \lambda_{i}}D_{i}\left(N\right)+D_{i}\left(t+1\right)\right)\times OP\times \phi \times E_{tran}$$
where OP denotes the operational period/phase, ϕ indicates the traffic (data generated) rate, and $E_{rec}$ and $E_{tran}$ represent the receiving energy and transmission energy, respectively. $D_{i}\left(N\right)$ denotes the degree of node *i* at time step *N*. By substituting Equations (Equation 9)–(Equation 11) in Equation (Equation 8), it can be rewritten as the following.
(12)$$\begin{array}{c} E_{i}\left(t+1\right)=(\left(\left(\sum_{N\in \lambda_{i}}D_{i}\left(N\right)+D_{i}\left(t+1\right)\right)\times OP\times \phi \right)\times \\ \left(E_{rec}+E_{tran}\right))+\left(\left|\Delta \right|\times OP\times \phi \times E_{tran}\right) \end{array}$$

It can also be reduced to the following equation,
(13)$$\begin{array}{c} E_{i}\left(t+1\right)=(OP\times \phi \times (\left(\sum_{N\in \lambda_{i}}D_{i}\left(N\right)+D_{i}\left(t+1\right)\right)\times \\ \left(E_{rec}+E_{tran}\right)+\left(|\Delta \right|\times E_{tran})) \end{array}$$
while the expected energy, Ex(λ,i), is obtained by
(14)$$Ex\left(\lambda ,i\right)=Degree\left(i\right)\times \left(\lambda +1\right)\times OP\times \phi \times E_{rec}$$
where OP denotes the operational period/phase, ϕ indicates the traffic (data generated) rate, and $E_{rec}$ signifies the receiving energy.

In the proposed cross-layer protocol, the sensor degree to elect a CH is applied through a timer-based. Following this principle, each sensor can set a timer upon starting its scheduling process (i.e., its own time), which should be different than its 1-hop neighboring nodes. The timer of each sensor node depends on its priority value, which has an inversely proportional relation with its degree, distance, and energy. A sensor that has higher degrees (1-hop neighbors), higher energy, and lower distance to the sink, within its neighborhood, should be able to begin its schedule process sooner than other nodes. Conversely, when a sensor has fewer degrees, lower energy, and higher distance to the sink, it should be delayed for a longer time to start its scheduling process. The timer for each sensor, $T_{node}$, is given by
(15)$$T_{node}=\left(1-P\left(\lambda ,i\right)\right)\times T_{sch},$$
where $T_{sch}$ is the scheduling interval set prior as a constant value for all sensor nodes during the deployment process. It should be noted that the priority value of each node P(λ,i) is between 0 and 1, where 0 means it has the highest chance to be a CH and 1 denotes it has the lowest chance to be a CH.

### 4.4. AUV Discovery Phase

The cluster heads are selected during the Distributed Clustering Phase and AUV has no prior information about them. During the Discovery phase, AUV travels on a predefined path (e.g., mowing-the-lawn pattern) and broadcasts small beacon packets to discover the CHs (e.g., polling the environment). Then, a nearby CH will reply by sending a small beacon to the approaching AUV. The CH (with the AUV current location) will then be added to the AUV list. Algorithms 1 and 2 detail the operation process between AUV and CH during the discovery phase.
**Algorithm 1:** AUV discovering algorithm
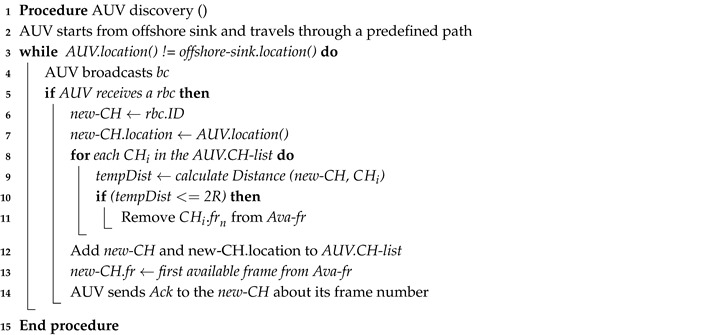




**Algorithm 2:** Cluster head operation during the discovery phase

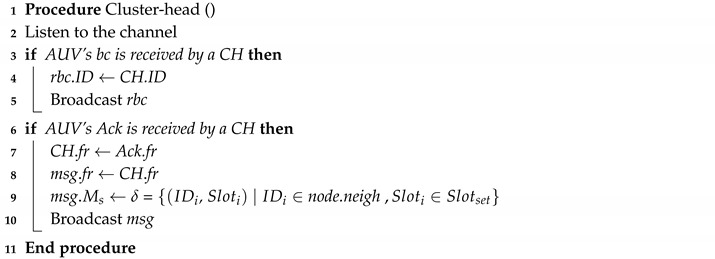




### 4.5. Normal Operational Phase

We illustrate the normal operational phase from two aspects: AUV data gathering and intracluster communication.

#### 4.5.1. Auv Data Gathering

In the AUV data gathering mission, the objective is to visit all CHs in the underwater environment in the shortest possible tour. This problem can be formulated into a well-known problem called the Traveling Salesman Problem (TSP) which is an NP-hard problem [22]. A greedy heuristic approach is adopted from our previous work [22] to obtain the near-optimal solutions in less computational time.

The 2-opt algorithm is also used to incrementally improve the initial tour by converting it to a tour without any crossed line. This algorithm swaps all possible pairs of edges while the current solution is replaced by the new one if the new tour is shortened [22]. This procedure will be stopped if no further improvement can be found.

Data packets are collected by each CH in a fixed rate and this procedure is independent of the AUV activity. After the tour planning, the AUV initiates its tour to collect the aggregated data. When AUV arrives, the CH transfers the buffered data to the AUV.

Upon the arrival of the AUV, a control packet is sent by the AUV to inform the approaching CH. All cluster members will be suspended after receiving the CH notification packet regarding the AUV arrival. Then, CH starts to transfer the data packets to the AUV. This procedure is repeated for the next CH within the tour until all CHs are visited. AUV finally returns to the sink to upload the collected information. This procedure has been shown in Figure 2. The next round of data gathering is similarly repeated.

#### 4.5.2. Intracluster Communications

The operational phase consists of several rounds, each of which includes two frames. Each frame includes several slots which are assigned for the data transmission or reception. In the operational phase, in periodic time slots, sensors become awake to transmit or receive data packets in some slots while they are in a sleeping mood in remaining slots. It means that sensors stay asleep when there is no data to be sent or received.

The purpose of having two frames in each round is to allow neighboring CHs assigning different frames (i.e., transmission time), as shown in Figure 3. During the second phase, CHs have already reserved the frames in a way that adjacent clusters have been assigned with different frames. This is because it guarantees that no collision occurs between members of different clusters. Each cluster member also reserves a specific time slot during the distributed clustering phase in a way that there would be no conflict with other members within the same cluster. Thus, sensors in different clusters with the same frame and slot can simultaneously transmit their packets without collisions. The MAC challenges and issues, such as hidden or exposed node problem, spatial-temporal uncertainty, and the near-far effect, are therefore addressed in CLMD [29].

After obtaining the frame and slot information from the initial two phases, each sensor knows its operational times as well as neighboring transmitting times. Each round repeats this pattern until further changes are required by the central scheduler due to energy depletion of CHs, or topology changes as sensors may move with the water current (if they are not anchored). Thus, the operational phase length depends on the number of rounds before rescheduling.

The offered traffic, λ, indicates the number of sent packets per second (p/s) for each node. Assuming the offered load is identical for all nodes, the length of each round, $T_{R}$, can be calculated by reversing the offered traffic. Thus, the length of $T_{R}$ is given by $\left(T_{R}=\frac{1}{\lambda }\right)$. The shorter $T_{R}$ is obtained with a higher λ and vice versa. A longer $T_{R}$ can create more opportunity for nodes to sleep. $T_{R}$ is divided by two to include two equal-size frames as $\left(T_{F}=\frac{T_{R}}{2}\right)$ and each $T_{F}$ contains a number of slots, $N_{s}$, with the same size. Let $D_{max}$ be the maximum node degree found within the whole network. The value of $D_{max}$ depends on the network topology and density. The number of slots can be given by $\left(N_{s}=D_{max}\right)$ to have enough time slots for the members of each cluster. The length of a slot, $T_{S}$ is calculated by
(16)$$T_{S}=\frac{T_{F}}{D_{max}},$$
while:(17)$$\;T_{S}\geq T_{delay}+G_{t}.$$

The slot length should be long enough to compensate for the high propagation delay in UWSNs. A small guard time, $G_{t}$, needs to be considered within the slot time to obtain zero packet loss due to the possibility of any conflict among sensors. The propagation delay, $T_{delay}$, is calculated by $\left(T_{delay}=\frac{R_{tr}}{V_{s}}\right)$, where $V_{s}$ and $R_{tr}$ are the velocity of sound in water and the transmission range of each sensor, respectively.

## 5. Experimental Results

In this section, we first discuss the simulation set-up of our protocol in the Aqua-Sim underwater simulation [31]. We then evaluate the most important metrics such as packet deliver ratio (PDR), network lifetime, goodput, energy consumption, and probability of collision. We also compare our proposed protocol with another cross-layer protocol (i.e., integrating the routing protocol in [6] with the MAC protocol in [32]). We finally present and analyze the simulation evaluation.

### 5.1. Simulation Setups

This study implemented CLMD in Aqua-Sim, an NS-2 based simulator for underwater sensor networks [31]. Simulations were performed with the following parameters, unless otherwise noted. The underwater communication channel described in Section 3 is used in our simulation. The sending, receiving, and idle powers are equal to 50 W, 0.158 W, and 0.008 W, respectively. The receiving power threshold and the transmission power are set to 10 dBreμPa, and 105 dBreμPa, respectively. Each sensor has initial energy equals to $E_{0}=40,000J$. The data packet size is equal to 1024 bits while the bit rate is set to 10 kbps. The packet generation rate is 0.01 packet per second.

The transmission range of each sensor is set to 100 m. The number of sensors ranges from 100 to 500, and sensors are uniformly deployed in a 1000 m × 1000 m field with a fixed depth at 300 m. There is one static sink at the corner of the network topology with (0, 500, 0) coordinates. The AUV travels at a depth of 250 m while its speed is 4 m/s. The simulation parameters to evaluate our model are summarized in Table 1.

The neighbor discovery phase is set to 60 s and the distributed clustering phase interval is set to 80 s while $T_{d}$ and $G_{t}$ are considered as 0.067 s each. The AUV discovery phase interval depends on the network dimensions, the predefined path and the AUV speed (during the discovery phase). In our simulation set-up, the AUV discovery time is 600 s. The normal operational phase is set to 7200 s. At the end of each operational phase, all phases are repeated to reschedule the sensors and increase network efficiency. Thus, the number of repetitions depends on the total simulation time and each phase interval. The energy of each phase is captured by the simulator considering the number of transmissions and receptions. The simulation time is set to 12 h while all the presented results are averaged over 50 runs for randomly generated topologies. The network lifetime is calculated by letting the simulation continues until the first sensor runs out of the energy.

### 5.2. Performance Metrics

We use and define the most important metrics for the performance evaluation of our proposed cross-layer protocol.

Packet Delivery Ratio (PDR) is defined as the ratio of the number of packets successfully received to the total number of packets generated across the network. Goodput is referred to the total amount of data packets successfully transmitted by sensors in the network within a given period of time (simulation time). Probability of collision is given by the following equation,
(18)$$P_{col}=1-\prod_{i=1}^{N-1}\frac{T_{s}-i\left(T_{d}+G_{t}\right)}{T_{s}}.$$
where *N* denotes the number of sensor nodes deployed in the network and $T_{s}$ indicates the scheduling interval. $T_{d}$ is the delivery time of each packet and $G_{t}$ is the guard time which is used to ensure that a packet is entirely received at the destination before starting of data transmission by another sensor. In our simulation set-up, we consider both $T_{d}$ and $G_{t}$ as 0.067 s each. This value is exactly equal to the maximum propagation delay of a transmitted packet.

Energy consumption shows the average energy consumed per packet successfully delivered. Network lifetime defines as the time when the first sensor runs out of energy.

### 5.3. Simulation Evaluation

In this section, we first evaluate the impact of routing layer on our propose cross-layer protocol. We also assess the impact of MAC layer on the cross-layer protocol.

#### 5.3.1. the Impact of Routing Layer on the Cross-Layer Protocol

The impact of the routing layer on the CLMD, CMDG, AEERP, and AUV_PN is evaluated in this section. Simulation parameters are fixed while the number of acoustic sensors varies from 100 to 500. Figure 4, Figure 5 and Figure 6 shows the simulation results for the packet delivery ratio, energy per packet, and network lifetime, respectively.

The packet delivery ratio for various network densities has been shown in Figure 4. CLMD has delivered a higher number of data packets to the destination compared to other similar schemes because of using a cross-layer design to limit the number of collisions. However, CMDG only uses a CSMA MAC protocol which is not optimal to be used in such a complex network. In the case of AUV_PN, no relay hop bound is considered to limit the number of hops between the CH and its members while it increases the chance of packet failure by transferring the packets over a longer distance. On the other hand, in AEERP, the chance of void occurrence between CH and its member increases when the network is sparse. Thus, in sparse scenarios, AEERP suffers from the lowest packet delivery ratio.

The energy efficiency of the models has been shown in Figure 5. CLMD consumes less energy because of bounding the packet transmissions to one hop and also reducing the number of collision by using an efficient MAC protocol.

The CMDG consumes higher energy than CLMD because it does not benefit from a cross-layer design like CLMD to reduce the number of collisions. The energy consumption in AUV_PN is higher than CLMD because of using CHs only for obtaining the list of PNs and not involving them in data collection. In this way, the number of nodes involving in data collection is lower than the actual value. In sparse networks, the void problem in AEERP increases energy consumption while the lack of relay hop bound in dense networks still wastes the energy.

The network lifetime of the proposed schemes has been shown in Figure 6. CLMD has higher network lifetime because of assigning sensors to a higher number of CHs in a uniform way, and the packet retransmission is highly reduced by utilizing an efficient MAC protocol. In CMDG, the network lifetime is higher than CLMD because of higher number of packet retransmissions due to using a CSMA MAC protocol. AUV_PN has a lower number of clusters which reduces the network lifetime. There is also no uniform assignment to the CHs in AEERP which can result in a shorter lifetime.

The latency of the sensor networks may be crucial in some applications. The latency depends on the tour length. The tour length also depends on different factors such as single-hop or multi-hop relay strategies, fixed tour or an adaptive tour, energy constraints, and reliability. In AEERP, the AUV travels a short path which is fixed; however, it increases the energy consumption and packet failure. The tour constructed by AUV_PN is not optimal as it crosses over itself. Sensor members in AUV_PN also need to transmit packets to the CH with a higher power to decrease the tour length. The tours established by CMDG and CLMD schemes are optimized using the 2-opt algorithm. In order to conduct a fair comparison for the time-critical applications, all these parameters need to be considered.

In the case of an emergency, there are several ways to decrease the latency. One way is to form the cluster heads with multi-hop neighbors (instead of one-hop neighboring nodes). In this way, the number of cluster heads is reduced and subsequently the AUV tour is shortened. However, there is a trade-off between latency and energy. If the latency is decreased by aggregating the data in fewer CHs, it will increase the energy consumption because the packets should be relayed in multi-hops.

#### 5.3.2. The Impact of Mac Layer on the Cross-Layer Protocol

In this set of simulation scenarios, the impact of sensor density on the performance of CLMD with DL-MAC, CLMD with ED-MAC, and CLMD with UWAN-MAC are examined. The number of sensors varies from 50 to 300, while other parameters are fixed. The simulation results for the packet delivery ratio, energy per packet, and network lifetime are shown in Figure 7, Figure 8 and Figure 9, respectively.

Figure 7 illustrates the PDR as a function of number of nodes. In this figure, CLMD with DL-MAC outperforms all other protocols by almost handling all data packets up until 100 nodes. When the node density further increases, the PDR of CLMD with DL-MAC slightly decreases by almost 1%. However, when the node density is up to 100 nodes, CLMD with ED-MAC delivers most of the packets (almost 98%), while it has a better delivery ratio than CLMD with UWAN-MAC protocol. When the node density is increased, the PDR of CLMD with ED-MAC decreases correspondingly to only deliver less than 78% in dense networks. This is mainly because of the spatial reuse property that is used in CLMD with ED-MAC is not as efficient as it is in CLMD with DL-MAC protocol.

It is noteworthy that the PDR of CLMD with UWAN-MAC with only 50 nodes is able to handle 99% of the data packets. However, when the number of nodes is further increased, the PDR of CLMD with UWAN-MAC reduces significantly by delivering approximately 20% in a high density network (with 300 nodes). This is mainly because of the long propagation delays, which cannot prevent the spatial-temporal uncertainty problem, leading to increasing the number of collisions.

As shown in Figure 8, the network goodput of all protocols is proportional to the nodes density. When the number of nodes is low (with 50 nodes), the network goodput increases correspondingly, except CLMD with UWAN-MAC, which reaches a saturation point within 150 nodes. Our proposed protocol, CLMD with DL-MAC, achieves a higher network goodput than that of CLMD with ED-MAC and CLMD with UWAN-MAC when their nodes density are the same. This is mainly because of that the higher the number of subframes in CLMD with DL-MAC, the more available slots can be settled as the nodes density increases. This means that CLMD with DL-MAC has the ability to handle more packets by having more slots to the sensors.

On the other hand, the goodput of CLMD with ED-MAC is continuously increased while the density of the nodes increased too. However, it still reaches a lower value than that of our proposed protocol. This is due to the simplicity of CLMD with ED-MAC during all three operational phases than others.

In Figure 9, we can observe that the higher the density of nodes, the higher the probability of collision which are presented to reserving the channel, except CLMD with DL-MAC protocol. Our proposed protocol (CLMD with DL-MAC) is more efficient than other protocols by having less percentage of packets lost when the density of nodes increases. This is due to the MAC protocol that is used in our proposed protocol, which is efficiently able to schedule the nodes by dividing the network area into a multi-layer to avoid any possible vertical collisions among the sensor nodes located in adjacent layers. Subsequently, each layer has *k* subframes to be settled by cluster heads located in the two-way distance from each other to avoid any potential horizontal conflict. By having a higher *k* subframe configurations with a fixed traffic rate, our proposed protocol is efficiently able to handle more nodes. However, the CLMD with UWAN-MAC begins to lose a small number of packets when the number of nodes is 50. When the number of nodes increases, the percentage of packets lost of CLMD with UWAN-MAC is significantly increased by losing approximately 80% of the total packets when the number of nodes is 300. This is mainly because of its inefficient scheduling, which significantly leads to overhearing and more collisions.

## 6. Conclusions

This paper has presented a study of a cross-layer protocol with the aim of developing efficient and scalable protocol to enhance the performance of underwater sensor networks (UWSNs). As UWSNs use acoustic waves, which are significantly different from terrestrial sensor networks that rely on radio waves to communicate with each other, designing a cross-layer protocol for underwater networks using acoustic communication is affected by many factors, such as high propagation delay, low bandwidth, and energy constraints. In this paper, a new cross-layer mobile data gathering scheme, CLMD, has been proposed to integrate the routing and MAC layer in UWSNs. In a distributed manner, CLMD selects a group of sensors as CHs and assigns a time slot to their members to facilitate their communication with cluster heads. The cluster heads have the responsibility to collect the information from sensors and deliver them to the AUV when it arrives. After clustering, the AUV travels the whole area to find the list of CHs and assigning a frame time to each cluster. If the the number of data packets is increased, CLMD still can keep its performance and the network lifetime high for a long time. CLMD can improve the scalability and resolve the void problem in sparse networks. We have evaluated the impact of routing layer on our propose cross-layer protocol. We then assessed the impact of MAC layer on the cross-layer protocol in order to select the best MAC protocol (CLMD with DL-MAC) for our proposed cross-layer protocol.

Using an extensive simulation study, the performance of CLMD has been compared against those of other cross-layer protocols recently reported in the literature. Simulation results have shown that CLMD outperforms other protocols in terms of packet deliver ratio, energy consumption, and network lifetime with varying numbers of nodes. Meanwhile, the performance of the chosen MAC protocols (CLMD with DL-MAC) has also been compared against those of other MAC protocols. The simulation results indicated that CLMD with DL-MAC can obtain a better delivery ratio, energy saving, and the probability of collision.

For the future work, we plan to enhance the performance of the cross-layer protocol by using an adaptive scheduling phase, which considers a changeable traffic rate during the operational phase. Another future interest lies in testing the reliability of real-life experiments. Although it is an expensive and complicated task to conduct a test bed in real experiments considering the different network sizes and scenarios involved, a real-life experiment with these ideas would better characterize the implementation outcomes of the different solutions proposed in this paper. It is also planned to investigate the low latency AUV-based data gathering schemes for time-critical applications in UWSNs.

## Figures and Tables

**Figure 1 sensors-20-04813-f001:**
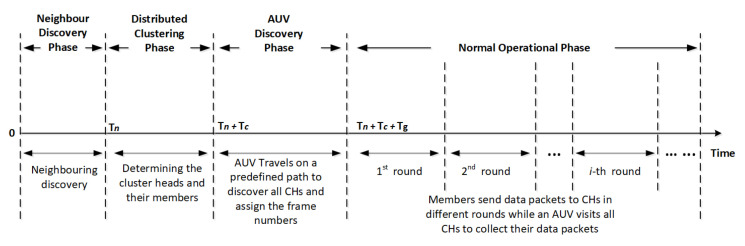
Timeline of Cross-Layer Mobile Data gathering (CLMD) protocol.

**Figure 2 sensors-20-04813-f002:**
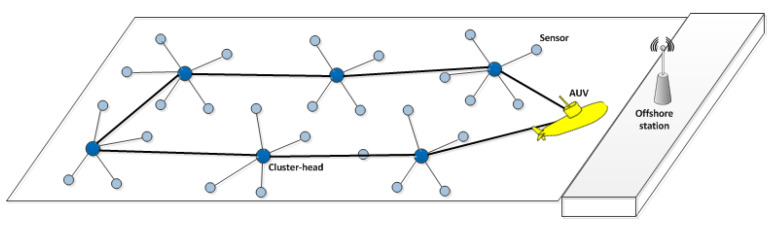
How the autonomous underwater vehicle (AUV) discovers cluster head and assigns different frames for each.

**Figure 3 sensors-20-04813-f003:**
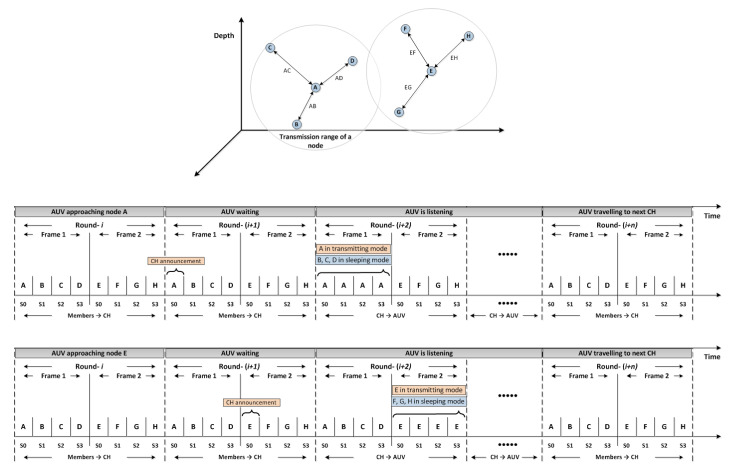
Timeline of our cross-layer proposed protocol.

**Figure 4 sensors-20-04813-f004:**
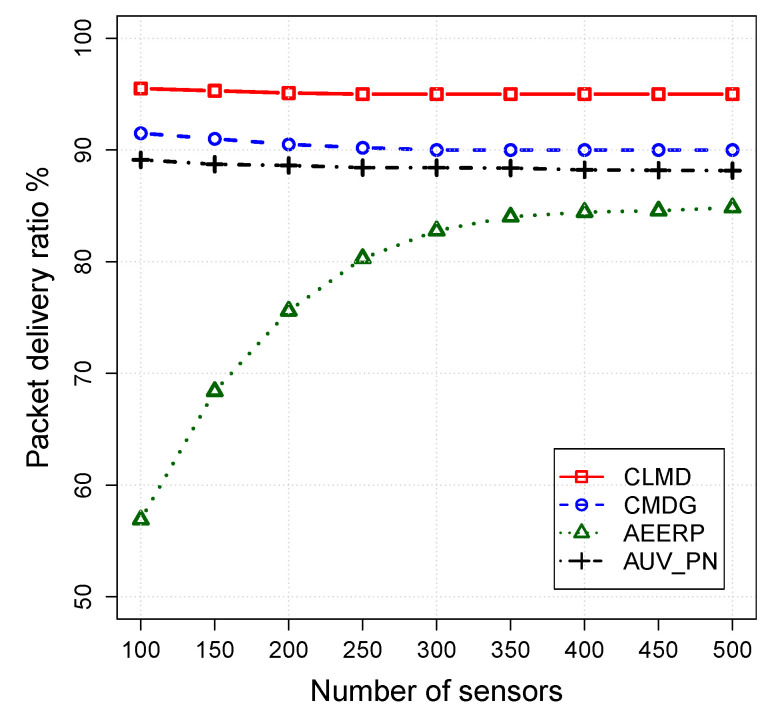
Packet delivery ratio vs. node density.

**Figure 5 sensors-20-04813-f005:**
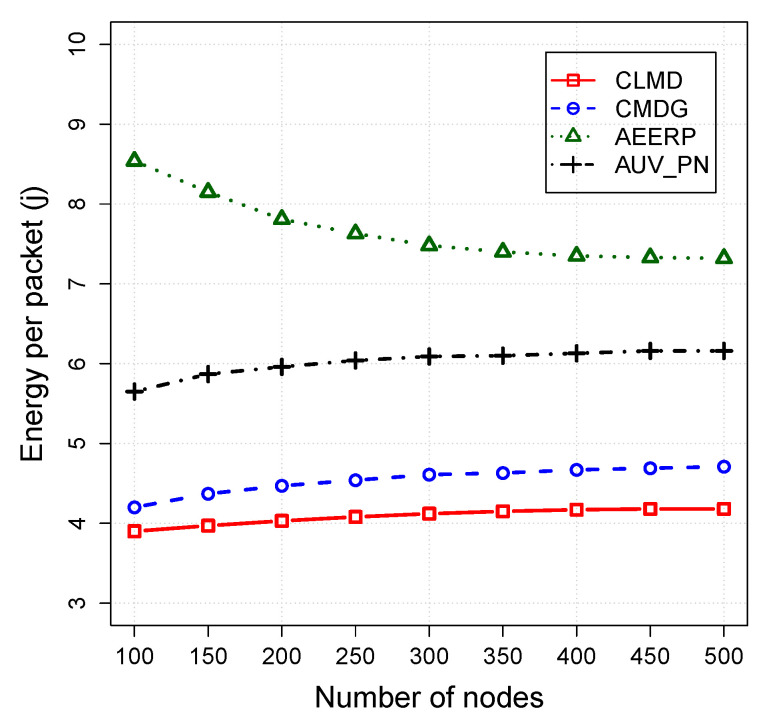
Energy consumption vs. node density.

**Figure 6 sensors-20-04813-f006:**
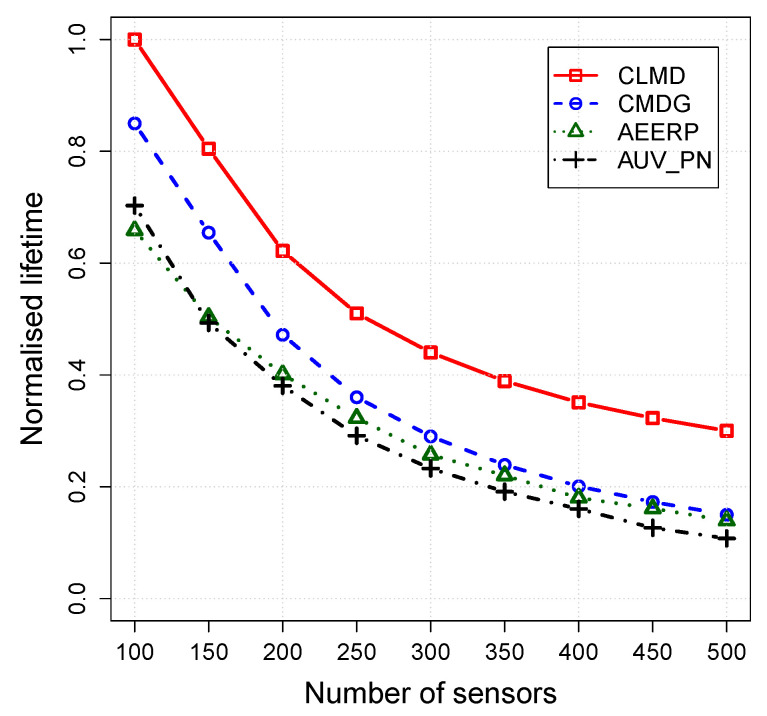
Network lifetime vs. node density.

**Figure 7 sensors-20-04813-f007:**
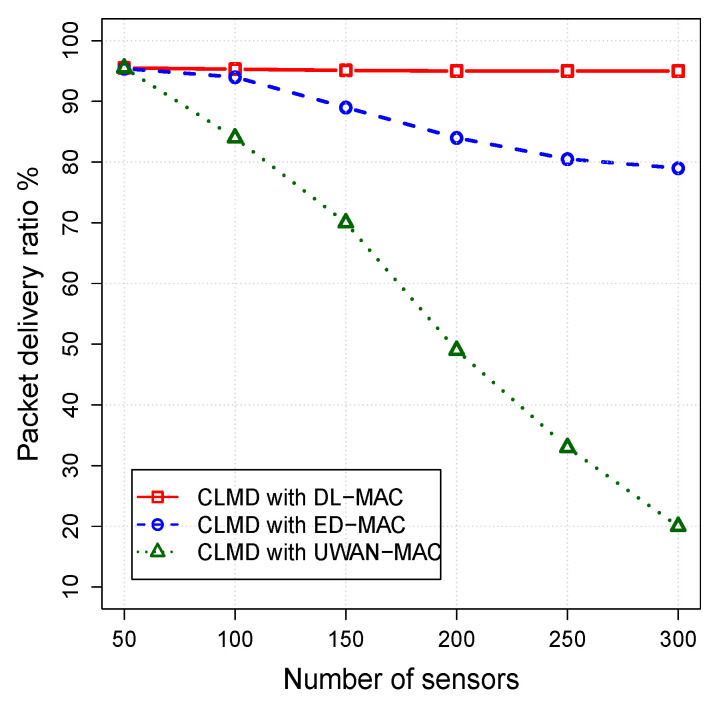
Packet delivery ratio vs. node density.

**Figure 8 sensors-20-04813-f008:**
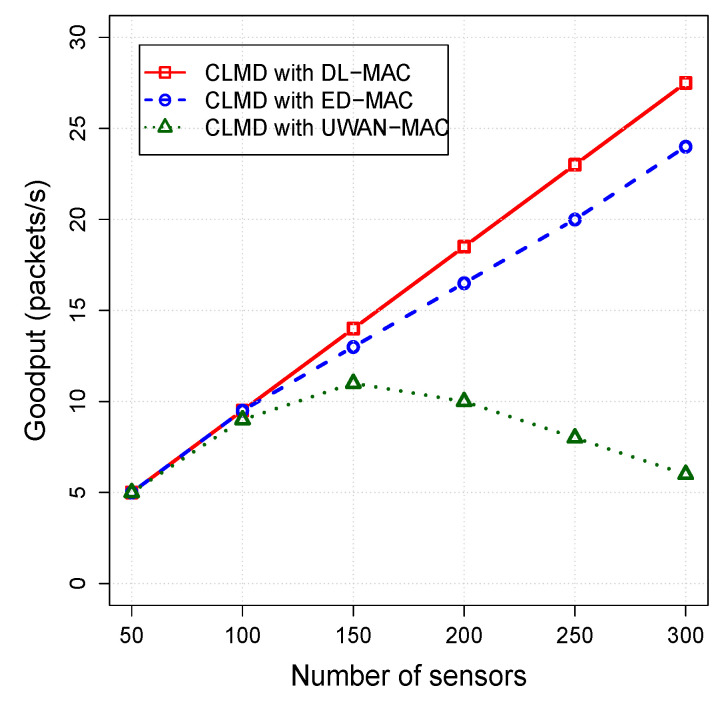
Goodput vs. node density.

**Figure 9 sensors-20-04813-f009:**
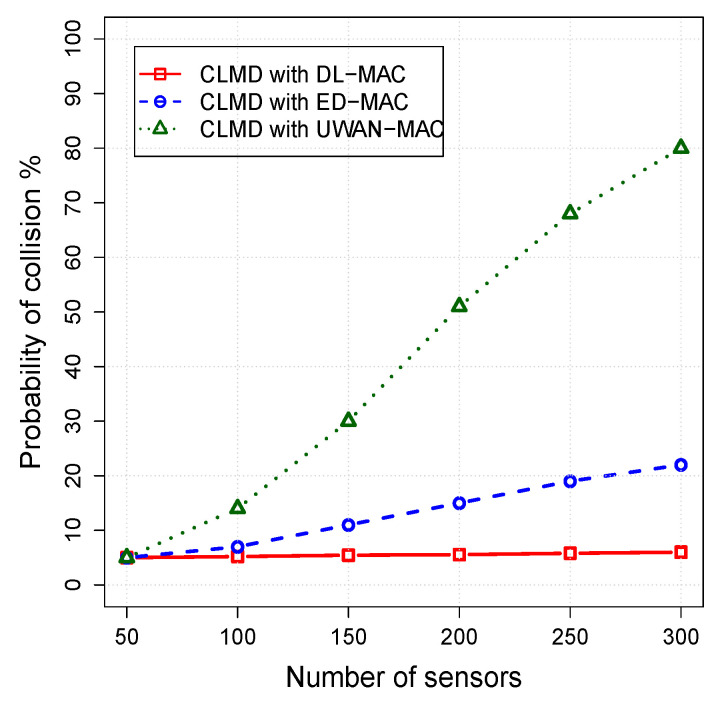
Probability of collision vs. node density.

**Table 1 sensors-20-04813-t001:** Simulation parameters.

Parameter	Value
Acoustic propagation speed	1500 m/s
Transmission power	105 dB re μ Pa
Receiving power threshold	10 dB re μ Pa
Sending power	50 W
Receiving power	0.158 W
Idle power	0.008 W
Transmission range	100 m
Packet generation rate	Every 100 s
Bit rate	10 kbps
Node number	100–500
Deployment region	1000 * 1000 * 300
AUV speed	4 m/s
AUV movement depth	250 m
Sink coordinate	(0, 500, 0)
Data packet size	1024 bits
Neighbour discovery interval	60 s
Distributed clustering interval	80 s
AUV discovery interval	600 s
Normal operational interval	7200 s
Initial energy of each node	40,000 J
$T_{d}$	0.067 s
$G_{t}$	0.067 s
Simulation time for each run	12 h

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
