# Peer review of "An AUV-Aided Cross-Layer Mobile Data Gathering Protocol for Underwater Sensor Networksâ€"

_sensors, 2020, doi:10.3390/s20174813_

Round 1
Reviewer 1 Report
Authors propose a novel Cross-Layer Mobile Data gathering scheme for Underwater Sensor Networks which uses the interaction between the MAC and routing layers. They use an autonomous Underwater Vehicle to periodically visit a group of clusters which are responsible for data collection from members.
The paper is interesting and the research contribution is good. There are some minor issues that should be fixed.
The related work section is weak. Only 3 related papers are discussed. There are many. At least authors should cite some surveys like this one:
Underwater wireless ad-hoc networks: A survey, Mobile ad hoc networks: Current status and future trends, 379-411, 2011
Underwater sensor nodes and networks, Sensors 13 (9), 11782-11796, 2013
Authors should include a flow chart diagram explaining the whole system procedure.
Authors should provide the reference to Aqua-Sim simulator.
There are 11 selfcitations to “Shahrabi, A. and Ghoreyshi, S.M.”. It is too much. More than 3-4 selfcitations for 30 references can not be allowed. I recommend rejecting the paper if there are more than 3-4 selcitations for such amount of references.
Author Response
We are very grateful to the reviewer for his thorough reviews of our paper. His comments have made it possible for us to significantly improve the present manuscript. Below are the reviewer’s recommendations, which were outlined in the Review Report Form, along with our response to each recommendation. Each comment is outlined one by one as they appeared in the Review Report Form, followed by a description of how we have addressed those comments in the paper.
Reply to Review (Reviewer 1):
Comment 1: The related work section is weak. Only 3 related papers are discussed. There are many. At least authors should cite some surveys like this one:
Response: Done. We have amended and updated the paper according to the recent comment. All the suggested references have been used along with four references more, which are as follows:
- Underwater wireless ad-hoc networks: A survey, Mobile ad hoc networks: Current status and future trends, 379-411, 2011
- Underwater sensor nodes and networks, Sensors 13 (9), 11782-11796, 2013
- Climent S, Sanchez A, Capella JV, Meratnia N, Serrano JJ. Underwater acoustic wireless sensor networks: advances and future trends in physical, MAC and routing layers. Sensors. 2014 Jan;14(1):795-833.
- Han S, Noh Y, Lee U, Gerla M. Optical-acoustic hybrid network toward real-time video streaming for mobile underwater sensors. Ad Hoc Networks. 2019 Feb 1;83:1-7.
- Madan R, Cui S, Lall S, Goldsmith NA. Cross-layer design for lifetime maximization in interference-limited wireless sensor networks. IEEE Transactions on Wireless Communications. 2006 Dec 19;5(11):3142-52.
- Wang H, Yang Y, Ma M, He J, Wang X. Network lifetime maximization with cross-layer design in wireless sensor networks. IEEE Transactions on Wireless Communications. 2008 Oct 28;7(10):3759-68.
Comment 2: Authors should include a flow chart diagram explaining the whole system procedure.
Response: Done. A timeline diagram has been added in page 4, which is explaining the whole system procedure or the whole process of our cross-layer protocol.
Comment 3: Authors should provide the reference to Aqua-Sim simulator.
Response: Done. The reference for Aqua-Sim Simulator has been provided in Section 5 (Experimental Results) in page 8, reference 29.
Comment 4: There are 11 self-citations to “Shahrabi, A. and Ghoreyshi, S.M.”. It is too much. More than 3-4 self-citations for 30 references cannot be allowed. I recommend rejecting the paper if there are more than 3-4 self-citations for such amount of references.
Response: Done. Only 4 references have been provided in this paper including the conference that is used to extend this journal.
Reviewer 2 Report
The authors proposed a novel protocol for data gathering from underwater sensor networks by using autonomous underwater vehicle. While similar approaches are available in the literature, the authors prove in the article, that their solution is providing better results. The paper is generally well written and easy to understand, I have only some minor questions.
a) The protocol consists of several phases, like Neighbor Discovery Phase, Distributed Clustering Phase, AUV discovery, etc.
It is not clear how often do these phases occur (for example only a single time at the beginning of the simulation), how long do they take and how are these phases and their energy consumption are considered in the simulation.
b) The first pass of the AUV discovering algorithm is not clear. If I understand correctly, the Cluster heads are selected during the Distributed Clustering Phase. Can the AUV get any apriori information about the cluster heads, or it is visiting all potential sensor nodes to find the cluster heads?
c) The latency of the sensor networks may be crucial in some applications. Is there any information on how the latency differs in the case of the compared four algorithms?
d) If an emergency situation occurs, is it possible to decrease the latency?
Author Response
Response to the Reviewer’s comments
August 13, 2020
We are very grateful to the reviewer for his thorough reviews of our paper. His comments have made it possible for us to significantly improve the present manuscript. Below are the reviewer’s recommendations, which were outlined in the Review Report Form, along with our response to each recommendation. Each comment is outlined one by one as they appeared in the Review Report Form, followed by a description of how we have addressed those comments in the paper.
Reply to Review (Reviewer 1):
Comment 1: The related work section is weak. Only 3 related papers are discussed. There are many. At least authors should cite some surveys like this one:
Response: Done. We have amended and updated the paper according to the recent comment. All the suggested references have been used along with four references more, which are as follows:
- Underwater wireless ad-hoc networks: A survey, Mobile ad hoc networks: Current status and future trends, 379-411, 2011
- Underwater sensor nodes and networks, Sensors 13 (9), 11782-11796, 2013
- Climent S, Sanchez A, Capella JV, Meratnia N, Serrano JJ. Underwater acoustic wireless sensor networks: advances and future trends in physical, MAC and routing layers. Sensors. 2014 Jan;14(1):795-833.
- Han S, Noh Y, Lee U, Gerla M. Optical-acoustic hybrid network toward real-time video streaming for mobile underwater sensors. Ad Hoc Networks. 2019 Feb 1;83:1-7.
- Madan R, Cui S, Lall S, Goldsmith NA. Cross-layer design for lifetime maximization in interference-limited wireless sensor networks. IEEE Transactions on Wireless Communications. 2006 Dec 19;5(11):3142-52.
- Wang H, Yang Y, Ma M, He J, Wang X. Network lifetime maximization with cross-layer design in wireless sensor networks. IEEE Transactions on Wireless Communications. 2008 Oct 28;7(10):3759-68.
Comment 2: Authors should include a flow chart diagram explaining the whole system procedure.
Response: Done. A timeline diagram has been added in page 4, which is explaining the whole system procedure or the whole process of our cross-layer protocol.
Comment 3: Authors should provide the reference to Aqua-Sim simulator.
Response: Done. The reference for Aqua-Sim Simulator has been provided in Section 5 (Experimental Results) in page 8, reference 29.
Comment 4: There are 11 self-citations to “Shahrabi, A. and Ghoreyshi, S.M.”. It is too much. More than 3-4 self-citations for 30 references cannot be allowed. I recommend rejecting the paper if there are more than 3-4 self-citations for such amount of references.
Response: Done. Only 4 references have been provided in this paper including the conference that is used to extend this journal.
Reply to Review (Reviewer 2):
The authors proposed a novel protocol for data gathering from underwater sensor networks by using autonomous underwater vehicle. While similar approaches are available in the literature, the authors prove in the article, that their solution is providing better results. The paper is generally well written and easy to understand, I have only some minor questions.
Comment 1: The protocol consists of several phases, like Neighbor Discovery Phase, Distributed Clustering Phase, AUV discovery, etc.
It is not clear how often do these phases occur (for example only a single time at the beginning of the simulation), how long do they take and how are these phases and their energy consumption are considered in the simulation.
Response: All phases are repeated in a periodic way. The total simulation time is 12 hours (43200 seconds). The neighbour discovery phase is set to 60 seconds. The distributed clustering phase interval is set to 80 seconds. The AUV discovery phase interval depends on the network dimensions, the predefined path and the AUV speed (during the discovery phase). In our simulation setup, the AUV discovery time is 600 seconds. The normal operational phase is set to 7200 seconds. At the end of each operational phase, all phases are repeated to reschedule the sensors and increase the network efficiency. Thus, the number of repetitions depends on the total simulation time and each phase interval. The energy of each phase is captured by the simulator considering the number of transmissions and receptions.
Comment 2: The first pass of the AUV discovering algorithm is not clear. If I understand correctly, the Cluster heads are selected during the Distributed Clustering Phase. Can the AUV get any apriori information about the cluster heads, or it is visiting all potential sensor nodes to find the cluster heads?
Response: The cluster heads are selected during the Distributed Clustering Phase and AUV has no prior information about them. During the Discovery phase, AUV travels on a predefined path (e.g. mowing-the-lawn pattern) and broadcasts small beacon packets to discover the CHs (e.g. polling the environment). Then, a nearby CH will reply by sending a small beacon to the approaching AUV. The CH (with the AUV current location) will then be added to the AUV list (Algorithms 1 and 2).
Comment 3: The latency of the sensor networks may be crucial in some applications. Is there any information on how the latency differs in the case of the compared four algorithms?
Response: The latency depends on the tour length. The tour length is also depending on different factors such as single-hop or multi-hops relay strategies, fixed tour or an adaptive tour, energy constraints and reliability. In AEERP, the AUV travels a short path which is fixed; however, it increases the energy consumption and packet failure. The tour constructed by AUV_PN is not optimal as it crosses over itself. Sensor members in AUV_PN also need to transmit packets to the CH with higher power to decrease the tour length. The tour established by CMDG and CLMD schemes are optimised using the 2-opt algorithm. In order to do a fair comparison for the time-critical applications, all these parameters need to be considered. As a future work, it is planned to investigate the low latency AUV-based data gathering schemes for UWSNs.
Comment 4: If an emergency situation occurs, is it possible to decrease the latency?
Response: In the case of emergency, there are several ways to decrease the latency. One way is to form the cluster heads with multi-hop neighbours (instead of one-hop neighbouring nodes). In this way, the number of cluster heads is reduced and subsequently the AUV tour is shortened. However, there is a trade-off between the latency and energy. If the latency is decreased by aggregating the data in fewer CHs, it will increase the energy consumption because the packets should be relayed in multi-hops.

Round 2
Reviewer 1 Report
Authors have fixed all my comments. I recommend to accept the paper.